# Calibrated Structured Prediction

**Volodymyr Kuleshov**
Department of Computer Science
Stanford University
Stanford, CA 94305

**Percy Liang**
Department of Computer Science
Stanford University
Stanford, CA 94305

## Abstract

In user-facing applications, displaying calibrated confidence measures—probabilities that correspond to true frequency—can be as important as obtaining high accuracy. We are interested in calibration for structured prediction problems such as speech recognition, optical character recognition, and medical diagnosis. Structured prediction presents new challenges for calibration: the output space is large, and users may issue many types of probability queries (e.g., marginals) on the structured output. We extend the notion of calibration so as to handle various subtleties pertaining to the structured setting, and then provide a simple recalibration method that trains a binary classifier to predict probabilities of interest. We explore a range of features appropriate for structured recalibration, and demonstrate their efficacy on three real-world datasets.

## 1 Introduction

Applications such as speech recognition [1], medical diagnosis [2], optical character recognition [3], machine translation [4], and scene labeling [5] have two properties: (i) they are instances of structured prediction, where the predicted output is a complex structured object; and (ii) they are user-facing applications for which it is important to provide accurate estimates of confidence. This paper explores confidence estimation for structured prediction.

Central to this paper is the idea of *probability calibration* [6, 7, 8, 9], which is prominent in the meteorology [10] and econometrics [9] literature. Calibration requires that the probability that a system outputs for an event reflects the true frequency of that event: of the times that a system says that it will rain with probability 0.3, then 30% of the time, it should rain. In the context of structured prediction, we do not have a single event or a fixed set of events, but rather a multitude of events that depend on the input, corresponding to different conditional and marginal probabilities that one could ask of a structured prediction model. We must therefore extend the definition of calibration in a way that deals with the complexities that arise in the structured setting.

We also consider the practical question of building a system that outputs calibrated probabilities. We introduce a new framework for calibration in structured prediction, which involves defining probabilities of interest, and then training binary classifiers to predict these probabilities based on a set of features. Our framework generalizes current methods for binary and multiclass classification [11, 12, 13], which predict class probabilities based on a single feature, the uncalibrated prediction score. In structured prediction, the space of interesting probabilities and useful features is considerably richer. This motivates us to introduce a new concept of *events* as well as a range of new features—margin, pseudomargin—which have varying computational demands. We perform a thorough study of which features yield good calibration, and find that domain-general features are quite good for calibrating MAP and marginal estimates over three tasks—object recognition, optical character recognition, and scene understanding. Interestingly, features based on MAP inference alone can achieve good calibration on marginal probabilities (which can be more difficult to compute).

Figure 1: In the context of an OCR system, our framework augments the structured predictor with calibrated confidence measures for a set of *events*, e.g., whether the first letter is "*l*".

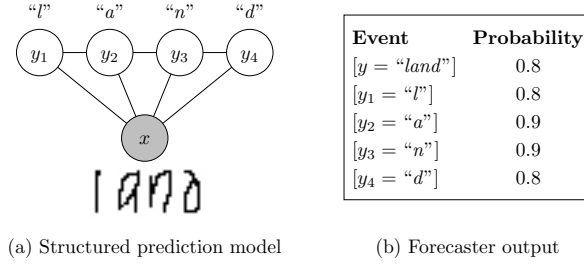

| Event | Probability |
|---|---|
| $[y = \text{``}land\text{''}]$ | 0.8 |
| $[y_1 = \text{``}l\text{''}]$ | 0.8 |
| $[y_2 = \text{``}a\text{''}]$ | 0.9 |
| $[y_3 = \text{``}n\text{''}]$ | 0.9 |
| $[y_4 = \text{``}d\text{''}]$ | 0.8 |

(a) Structured prediction model      (b) Forecaster output

## 2 Background

### 2.1 Structured Prediction

In structured prediction, we want to assign a structured label $y = (y_1, \ldots, y_L) \in \mathcal{Y}$ to an input $x \in \mathcal{X}$. For example, in optical character recognition (OCR), $x$ is a sequence of images and $y$ is the sequence of associated characters (see Figure 1(a)); note that the number of possible outputs $y$ for a given $x$ may be exponentially large.

A common approach to structured prediction is conditional random fields (CRFs), where we posit a probabilistic model $p_\theta(y \mid x)$. We train $p_\theta$ by optimizing a maximum-likelihood or a max-margin objective over a training set $\{(x^{(i)}, y^{(i)})\}_{i=1}^n$, assumed to be drawn i.i.d. from an unknown data-generating distribution $\mathbb{P}(x, y)$. The promise of a probabilistic model is that in addition to computing the most likely output $\hat{y} = \arg\max_y p_\theta(y \mid x)$, we can also get its probability $p_\theta(y = \hat{y} \mid x) \in [0, 1]$, or even marginal probabilities $p_\theta(y_1 = \hat{y}_1 \mid x) \in [0, 1]$.

### 2.2 Probabilistic Forecasting

Probabilities from a CRF $p_\theta$ are just numbers that sum to 1. In order for these probabilities to be useful as confidence measures, we would ideally like them to be *calibrated*. Calibration intuitively means that whenever a forecaster assigns 0.7 probability to an event, it should be the case that the event actually holds about 70% of the time. In the case of binary classification ($\mathcal{Y} = \{0, 1\}$), we say that a forecaster $F : \mathcal{X} \to [0, 1]$ is *perfectly calibrated* if for all possible probabilities $p \in [0, 1]$:

$$\mathbb{P}[y = 1 \mid F(x) = p] = p. \tag{1}$$

Calibration by itself does not guarantee a useful confidence measure. A forecaster that always outputs the marginal class probability $F(x) = \mathbb{P}(y = 1)$ is calibrated but useless for accurate prediction. Good forecasts must also be *sharp*, i.e., their probabilities should be close to 0 or 1.

**Calibration and sharpness.** Given a forecaster $F : \mathcal{X} \to [0, 1]$, define $T(x) = \mathbb{E}[y \mid F(x)]$ to be the true probability of $y = 1$ given a that $x$ received a forecast $F(x)$. We can use $T$ to decompose the $\ell_2$ prediction loss as follows:

$$\mathbb{E}[(y - F(x))^2] = \mathbb{E}[(y - T(x))^2] + \mathbb{E}[(T(x) - F(x))^2] \tag{2}$$

$$= \underbrace{\text{Var}[y]}_{\text{uncertainty}} - \underbrace{\text{Var}[T(x)]}_{\text{sharpness}} + \underbrace{\mathbb{E}[(T(x) - F(x))^2]}_{\text{calibration error}}. \tag{3}$$

The first equality follows because $y - T(x)$ has expectation 0 conditioned on $F(x)$, and the second equality follows from the variance decomposition of $y$ onto $F(x)$.

The three terms in (3) formalize our intuitions about calibration and sharpness [7]. The calibration term measures how close the predicted probability is to the true probability over that region and is a natural generalization of perfect calibration (1) (which corresponds to zero calibration error). The sharpness term measures how much variation there is in the true probability across forecasts. It does not depend on the numerical value of the forecaster $F(x)$, but only the induced grouping of points; it is maximized by making $F(x)$ closer to 0 and 1. Uncertainty does not depend on the forecaster and can be mostly ignored; note that it is always greater than sharpness and thus ensures that the loss stays positive.

**Examples.** To illustrate the difference between calibration error (lower is better) and sharpness (higher is better), consider the following binary clas-

| input $x$ | 0 | 1 | 2 | calib. | sharp. |
|---|---|---|---|---|---|
| true $\mathbb{P}(y \mid x)$ | 0 | 1 | 0.5 | 0 | 0.167 |
| calibrated, unsharp $p_\theta(y \mid x)$ | 0.5 | 0.5 | 0.5 | 0 | 0 |
| uncalibrated, sharp $p_\theta(y \mid x)$ | 0.2 | 0.8 | 0.4 | 0.03 | 0.167 |
| balanced $p_\theta(y \mid x)$ | 0 | 0.75 | 0.75 | 0 | 0.125 |

sification example: we have a uniform distribution ($\mathbb{P}(x) = 1/3$) over inputs $\mathcal{X} = \{0, 1, 2\}$. For $x \in \{0, 1\}$, $y = x$ with probability 1, and for $x = 2$, $y$ is either 0 or 1, each with probability $\frac{1}{2}$.

Setting $p_\theta(y \mid x) \equiv 0.5$ would achieve perfect calibration (0) but not sharpness (0). We can get excellent sharpness (0.167) but suffer in calibration (0.03) by predicting probabilities $0.2, 0.8, 0.4$. We can trade off some sharpness (0.125) for perfect calibration (0) by predicting 0 for $x = 0$ and 0.75 for $x \in \{1, 2\}$.

**Discretized probabilities.** We have assumed so far that the forecaster might return arbitrary probabilities in $[0, 1]$. In this case, we might need an infinite amount of data to estimate $T(x) = \mathbb{E}[y \mid F(x)]$ accurately for each value of $F(x)$. In order to estimate calibration and sharpness from finite data, we use a discretized version of calibration and sharpness. Let $\mathcal{B}$ be a partitioning of the interval $[0, 1]$; for example $\mathcal{B} = \{[0, 0.1), [0.1, 0.2), \dots \}$. Let $B : [0, 1] \to \mathcal{B}$ map a probability $p$ to the interval $B(p)$ containing $p$; e.g., $B(0.15) = [0.1, 0.2)$. In this case, we simply redefine $T(x)$ to be the true probability of $y = 1$ given that $F(x)$ lies in a bucket: $T(x) = \mathbb{E}[y \mid B(F(x))]$. It is not hard to see that discretized calibration estimates form an upper bound on the calibration error (3) [14].

## 3 Calibration in the Context of Structured Prediction

We have so far presented calibration in the context of binary classification. In this section, we extend these definitions to structured prediction. Our ultimate motivation is to construct forecasters that augment pre-trained structured models $p_\theta(y|x)$ with confidence estimates. Unlike in the multiclass setting [12], we cannot learn a forecaster $F_y : \mathcal{X} \to [0, 1]$ that targets $\mathbb{P}(y \mid x)$ for each $y \in \mathcal{Y}$ because the cardinality of $\mathcal{Y}$ is too large; in fact, the user will probably not be interested in every $y$.

**Events of interest.** Instead, we assume that for a given $x$ and associated prediction $y$, the user is interested in a set $\mathcal{I}(x)$ of *events* concerning $x$ and $y$. An event $E \in \mathcal{I}(x)$ is a subset $E \subseteq \mathcal{Y}$; we would like to determine the probability $\mathbb{P}(y \in E \mid x)$ for each $E \in \mathcal{I}(x)$. Here are two useful types of events that will serve as running examples:

1. $\{\text{MAP}(x)\}$, which encodes whether $\text{MAP}(x) = \arg \max_y p_\theta(y \mid x)$ is correct.
2. $\{y : y_j = \text{MAP}(x)_j\}$, which encodes whether the label at position $j$ in $\text{MAP}(x)$ is correct.

In the OCR example (Figure 1), suppose we predict $\text{MAP}(x) = $ "*land*". Define the events of interest to be the MAP and the marginals: $\mathcal{I}(x) = \{\{\text{MAP}(x)\}, \{y : y_1 = \text{MAP}(x)_1\}, \dots, \{y : y_L = \text{MAP}(x)_L\}\}$. Then we have $\mathcal{I}(x) = \{\{$"*land*"$\}, \{y : y_1 = $"*l*"$\}, \{y : y_2 = $"*a*"$\}, \{y : y_3 = $"*n*"$\}, \{y : y_4 = $"*d*"$\}\}$. Note that the events of interest $\mathcal{I}(x)$ depend on $x$ through $\text{MAP}(x)$.

**Event pooling.** We now define calibration in analogy with (1). We will construct a forecaster $F(x, E)$ that tries to predict $\mathbb{P}(y \in E \mid x)$. As we remarked earlier, we cannot make a statement that holds uniformly for all events $E$; we can only make a guarantee in expectation. Thus, let $E$ be drawn uniformly from $\mathcal{I}(x)$, so that $\mathbb{P}$ is extended to be a joint distribution over $(x, y, E)$. We say that a forecaster $F : \mathcal{X} \times 2^{\mathcal{Y}} \mapsto [0, 1]$ is perfectly calibrated if

$$\mathbb{P}(y \in E \mid F(x, E) = p) = p. \tag{4}$$

In other words, averaged over all $x, y$ and events of interest $E \in \mathcal{I}(x)$, whenever the forecaster outputs probability $p$, then the event $E$ actually holds with probability $p$. Note that this defini­tion corresponds to perfect binary calibration (1) for the transformed pair of variables $y' = \mathbb{I}[y \in E], x' = (x, E)$. As an example, if $\mathcal{I}(x) = \{\{\text{MAP}(x)\}\}$, then (4) says that of all the MAP predic­tions with confidence $p$, a $p$ fraction will be correct. If $\mathcal{I}(x) = \{\{y : y_j = \text{MAP}(x)_j\}\}_{j=1}^L$, then (4) states that out of all the marginals (pooled together across all samples $x$ and all positions $j$) with confidence $p$, a $p$ fraction will be correct.

**Algorithm 1** Recalibration procedure for calibrated structured prediction.

---
**Input:** Features $\phi(x, E)$ from trained model $p_\theta$, event set $\mathcal{I}(x)$, recalibration set $\mathcal{S} = \{(x_i, y_i)\}_{i=1}^n$.
**Output:** Forecaster $F(x, E)$.
Construct the events dataset: $\mathcal{S}_{\text{binary}} = \{(\phi(x, E), \mathbb{I}[y \in E]) : (x, y) \in \mathcal{S}, E \in \mathcal{I}(x)\}$
Train the forecaster $F$ (e.g., $k$-NN or decision trees) on $\mathcal{S}_{\text{binary}}$.

---

The second example hints at an important subtlety inherent to having multiple events in structured prediction. The confidence scores for marginals are only calibrated when averaged over all positions. If a user only looked at the marginals for the first position, she might be sorely disappointed. As an extreme example, suppose $y = (y_1, y_2)$ and $y_1$ is 0 or 1 with probability $\frac{1}{2}$ while $y_2 \equiv 1$. Then a forecaster that outputs a confidence of 0.75 for both events $\{y : y_1 = 1\}$ and $\{y : y_2 = 1\}$ will be perfectly calibrated. However, neither event is calibrated in isolation ($\mathbb{P}(y_1 = 1 \mid x) = \frac{1}{2}$ and $\mathbb{P}(y_2 = 1 \mid x) = 1$). Finally, perfect calibration can be relaxed; following (3), we may define the calibration error to be $\mathbb{E}[(T(x, E) - F(x, E))^2]$, where $T(x, E) \stackrel{\text{def}}{=} \mathbb{P}(y \in E \mid F(x, E))$.

# 4 Constructing Calibrated Forecasters

Having discussed the aspects of calibration specific to structured prediction, let us now turn to the problem of constructing calibrated (and sharp) forecasters from finite data.

**Recalibration framework.** We propose a framework that generalizes existing recalibration strategies to structured prediction models $p_\theta$. First, the user specifies a set of events of interest $\mathcal{I}(x)$ as well as *features* $\phi(x, E)$, which will in general depend on the trained model $p_\theta$. We then train a forecaster $F$ to predict whether the event $E$ holds (i.e. $\mathbb{I}[y \in E]$) given features $\phi(x, E)$. We train $F$ by minimizing the empirical $\ell_2$ loss over a recalibration set $\mathcal{S}$ (disjoint from the training examples): $\min_F \sum_{(x,y) \in \mathcal{S}} \sum_{E \in \mathcal{I}(x)} (F(x, E) - \mathbb{I}[y \in E])^2$. Algorithm 1 outlines our procedure.

As an example, consider again the OCR setting in Figure 1. The margin feature $\phi(x, E) = \log p_\theta(\text{MAP}^{(1)}(x)) - \log p_\theta(\text{MAP}^{(2)}(x))$ (where $\text{MAP}^{(1)}(x)$ and $\text{MAP}^{(2)}(x)$ are the first and second highest scoring labels for $x$ according to $p_\theta$, respectively) will typically correlate with the event that the MAP prediction is correct. We can perform isotonic regression using this feature on the recalibration set $\mathcal{S}$ to produce well-calibrated probabilities.

In the limit of infinite data, Algorithm 1 minimizes the expected loss $\mathbb{E}[(F(x, E) - \mathbb{I}[y \in E])^2]$, where the expectation is over $(x, y, E)$. By (3), the calibration error $\mathbb{E}[(T(x, E) - F(x, E))^2]$ will also be small. If there are not too many features $\phi$, we can drive the $\ell_2$ loss close to zero with a nonparametric method such as $k$-NN. This is also why isotonic regression is sensible for binary recalibration: we first project the data into a highly informative one-dimensional feature space; then we predict labels from that space to obtain small $\ell_2$ loss.

Note also that standard multiclass recalibration is a special case of this framework, where we use the raw uncalibrated score from $p_\theta$ as a single feature. In the structured setting, one must invest careful thought in the choice of classifier and features; we discuss these choices below.

**Features.** Calibration is possible even with a single constant feature (e.g. $\phi(x, E) \equiv 1$), but sharpness depends strongly on the features' quality. If $\phi$ collapses points of opposite labels, no forecaster will be able to separate them and be sharp. While we want informative features, we can only afford to have a few, since our recalibration set is typically small.

Compared to calibration for binary classification, our choice of features must also be informed by their computational requirements: the most informative features might require performing full inference in an intractable model. It is therefore useful to think of features as belonging to one of three types, depending on whether they are derived from unstructured classifiers (e.g. an SVM trained individually on each label), MAP inference, or marginal inference. In Section 5, we will show that marginal inference produces the sharpest features, but clever MAP-based features can do almost as well.

In Table 1, we propose several features that follow our guiding principles and that illustrate the computational tradeoffs inherent to structured prediction.

| | MAP recalibration on $y$ | | Marginal recalibration on $y_j$ | |
|---|---|---|---|---|
| *Type* | *Name* | *Definition* | *Name* | *Definition* |
| none | $\phi_1^{\mathrm{no}}$ : SVM margin | $\min_j \mathrm{mrg}_{y_j}[s_j^{\mathrm{SVM}}(y_j)]$ | $\phi_2^{\mathrm{no}}$ : SVM margin | $\mathrm{mrg}_{y_j}[s_j^{\mathrm{SVM}}(y_j)]$ |
| MAP | $\phi_1^{\mathrm{mp}}$ : Label length | $\lvert y^{\mathrm{MAP}}\rvert$ | $\phi_4^{\mathrm{mp}}$ : Label freq. | % positions $j'$ labeled $y_j^{\mathrm{MAP}}$ |
| | $\phi_2^{\mathrm{mp}}$ : Admissibility | $\mathbb{I}[y^{\mathrm{MAP}} \in \mathcal{G}(x)]$ | $\phi_5^{\mathrm{mp}}$ : Neighbors | % neighbors $j'$ labeled $y_j^{\mathrm{MAP}}$ |
| | $\phi_3^{\mathrm{mp}}$ : Margin | $\mathrm{mrg}_y[p_\theta(y \mid x)]$ | $\phi_6^{\mathrm{mp}}$ : Label type | $\mathbb{I}[y_j^{\mathrm{MAP}} \in \mathcal{L}(x)]$ |
| | | | $\phi_7^{\mathrm{mp}}$ : Pseudomargin | $\mathrm{mrg}_{y_j}[p_\theta(y_j \mid y_{-j}^{\mathrm{MAP}}, x)]$ |
| Marg. | $\phi_1^{\mathrm{mg}}$ : Margin | $\min_j \mathrm{mrg}_{y_j}[p_\theta(y_j \mid x)]$ | $\phi_2^{\mathrm{mg}}$ : Margin | $\mathrm{mrg}_{y_j}[p_\theta(y_j \mid x)]$ |
| | | | $\phi_3^{\mathrm{mg}}$ : Concordance | $\mathbb{I}[y_j^{\mathrm{MG}} = y_j^{\mathrm{MAP}}]$ |

Table 1: Features for MAP recalibration ($\mathcal{I}(x) = \{\{y^{\mathrm{MAP}}(x)\}\}$) and marginal recalibration ($\mathcal{I}(x) = \{\{y : y_j = y^{\mathrm{MAP}}(x)_j\}\}_{j=1}^L$). We consider three types of features, requiring either unstructured, MAP, or marginal inference. For a generic function $f$, define $\mathrm{mrg}_a f(a) \triangleq f(a^{(1)}) - f(a^{(2)})$, where $a^{(1)}$ and $a^{(2)}$ are the top two inputs to $f$, ordered by $f(a)$. Let $y_j^{\mathrm{MG}} \triangleq \arg\max_{y_j} p_\theta(y_j \mid x)$; let $s_j^{\mathrm{SVM}}(y_j)$ be the score of an SVM classifier predicting label $y_j$. Features $\phi_2^{\mathrm{mp}}$ and $\phi_6^{\mathrm{mp}}$ require domain-specific knowledge: defining admissible sets $\mathcal{G}(x), \mathcal{L}(x)$. In OCR, $\mathcal{G}$ are all English words and $\mathcal{L}(x)$ are similar-looking letters. Percentages in $\phi_4^{\mathrm{mp}}$ and $\phi_5^{\mathrm{mp}}$ are relative to all the labels in $y^{\mathrm{MAP}}$.

**Region-based forecasters.** Recall from (4) that calibration examines the true probability of an event ($y \in E$) *conditioned* on the forecaster's prediction $F(x, E) = p$. By limiting the number of different probabilities $p$ that $F$ can output, we can more accurately estimate the true probability for each $p$. To this end, let us partition the feature space (the range of $\phi$) into regions $\mathcal{R}$, and output a probability $F_R \in [0, 1]$ for each region $R \in \mathcal{R}$. Formally, we consider *region-based* forecasters of the form $F(x, E) = \sum_{R \in \mathcal{R}} F_R \mathbb{I}[\phi(x, E) \in R]$, where $F_R$ is the fraction of points in region $R$ (that is, $(x, E)$ for which $\phi(x, E) \in R$) for which the event holds ($y \in E$). Note that the partitioning $\mathcal{R}$ could itself depend on the recalibration set. Two examples of region-based forecasters are $k$-nearest neighbors ($k$-NN) and decision trees.

Let us obtain additional insight into the performance of region-based forecasters as a function of recalibration set size. Let $\mathcal{S}$ denote here a recalibration set of size $n$, which is used to derive a partitioning $\mathcal{R}$ and probability estimates $F_R$ for each region $R \in \mathcal{R}$. Let $T_R \triangleq \mathbb{P}(y \in E \mid \phi(x, E) \in R)$ be the true event probability for region $R$, and $w_R \triangleq \mathbb{P}(\phi(x, E) \in R)$ be the probability mass of region $R$. We may rewrite the expected calibration error (3) of $F_R$ trained on a random $\mathcal{S}$ of size $n$ (drawn i.i.d. from $\mathbb{P}$) as

$$\mathrm{CalibrationError}_n = \mathbb{E}_{\mathcal{R}} \left[ \sum_{R \in \mathcal{R}} w_R \mathbb{E}_{\mathcal{S}}[(F_R - T_R)^2 \mid \mathcal{R}] \right]. \tag{5}$$

We see that there is a classic bias-variance tradeoff between having smaller regions (lower bias, increased sharpness) and having more data points per region (lower variance, better calibration):

$$\mathbb{E}[(F_R - T_R)^2 \mid \mathcal{R}] = \underbrace{(\mathbb{E}[F_R \mid \mathcal{R}] - T_R)^2}_{\text{bias}} + \underbrace{\mathbb{E}[(F_R - \mathbb{E}[F_R \mid \mathcal{R}])^2 \mid \mathcal{R}]}_{\text{variance}}.$$

If $\mathcal{R}$ is a fixed partitioning independent of $\mathcal{S}$, then the bias will be zero, and the variance is due to an empirical average, falling off as $1/n$. However, both $k$-NN and decision trees produce biased estimates $F_R$ of $T_R$ because the regions are chosen adaptively, which is important for achieving sharpness. In this case, we can still ensure that the calibration error vanishes to zero if we let the regions grow uniformly larger: $\min_{R \in \mathcal{R}} |\{(x, y) \in \mathcal{S} : \phi(x, E) \in R, E \in \mathcal{I}(x)\}| \xrightarrow{P} \infty$.

## 5 Experiments

We test our proposed recalibrators and features on three real-world tasks.

**Multiclass image classification.** The task is to predict an image label given an image. This setting is a special case of structured prediction in which we show that our framework improves over existing multiclass recalibration strategies. We perform our experiments on the CIFAR-10 dataset [15],

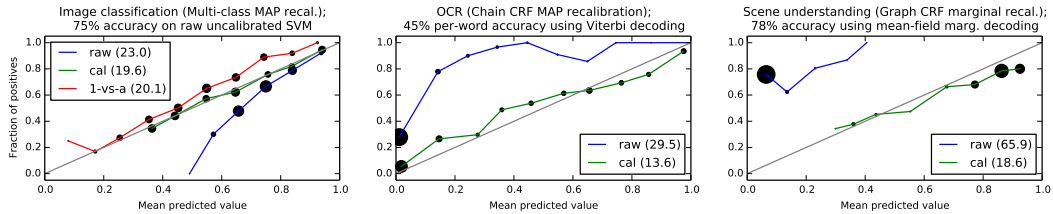

Figure 2: MAP recalibration in the multiclass and chain CRF settings (left and middle) and marginal recalibration of the graph CRF (right). The legend includes the $\ell_2$ loss before and after calibration. The radius of the black balls reflects the number of points having the given forecasted and true probabilities.

which consists of 60,000 32x32 color images of different types of animals and vehicles (ten classes in total). We train a linear SVM on features derived from $k$-means clustering and that produce high accuracies (79%) on this dataset [16]. We use 800 out of the 1600 features having the highest mutual information with the label (the drop in performance is negligible). 38,000 images were used for training, 2,000 for calibration, and 20,000 for testing.

**Optical character recognition.** The task is to predict the word (sequence of characters) given a sequence of images (Figure 1). Calibrated OCR systems can be useful for automatic sorting of mail. This setting demonstrates calibration on a tractable linear-chain CRF. We used a dataset consisting of $\sim$ 8-character-long words from 150 human subjects [3]. Each character is rasterized into a $16 \times 8$ binary image. We chose 2000 words for training and another 2000 for testing. The remaining words are subsampled in various ways to produce recalibration sets.

**Scene understanding.** Given an image divided into a set of regions, the task is to label each region with its type (e.g. person, tree, etc.). Calibrated scene understanding is important for building autonomous agents that try to take optimal actions in the environment, integrating over uncertainty. This is a structured prediction setting in which inference is intractable. We conduct experiments on a post-processed VOC Pascal dataset [5]. In brief, we train a graph CRF to predict the joint labeling $y_i$ of superpixels $y_{ij}$ in an image ($\sim$ 100 superpixels per image; 21 possible labels). The input $x_i$ consists of 21 node features; CRF edges connect adjacent superpixels. We use 600 examples for training, 500 for testing and subsample the remaining $\sim$ 800 examples to produce calibration sets. We perform MAP inference using AD3, a dual composition algorithm; we use a mean field approximation to compute marginals.

**Experimental setup.** We perform both MAP and marginal calibration as described in Section 3. We use decision trees and $k$-NN as our recalibration algorithms and examine the quality of our forecasts based on calibration and sharpness (Section 2). We further discretize probabilities into buckets of size 0.1: $\mathcal{B} = \{[\frac{i-1}{10}, \frac{i}{10}) : i = 1, \ldots, 10\}$.

We report results using *calibration curves*: For each test point $(x_i, E_i, y_i)$, let $f_i = F(x_i, E_i) \in [0, 1]$ be the forecasted probability and $t_i = \mathbb{I}[y_i \in E_i] \in \{0, 1\}$ be the true outcome. For each bucket $B \in \mathcal{B}$, we compute averages $f_B = N_B^{-1} \sum_{i: f_i \in B} f_i$ and $t_B = N_B^{-1} \sum_{i: f_i \in B} t_i$, where $N_B = |\{f_i \in B\}|$ is the number of points in bucket $B$. A calibration curve plots the $t_B$ as a function of $f_B$. Perfect calibration corresponds to a straight line. See Figure 2 for an example.

## 5.1 "Out-of-the-Box" Recalibration

We would first like to demonstrate that our approach works well "out of the box" with very simple parameters: a single feature, $k$-NN with $k = 100$, and a reasonably-sized calibration set. We report results in three settings: (i) multiclass and (ii) chain CRF MAP recalibration with the margin feature $\phi_1^{\text{mg}}$ (Figure 2, left, middle), as well as (iii) graph CRF marginal recalibration with the margin feature $\phi_2^{\text{mg}}$ (Figure 2, right). We use calibration sets of 2,000, 1,000, and 300 (respectively) and compare to the raw CRF probabilities $p_\theta(y \in E \mid x)$.

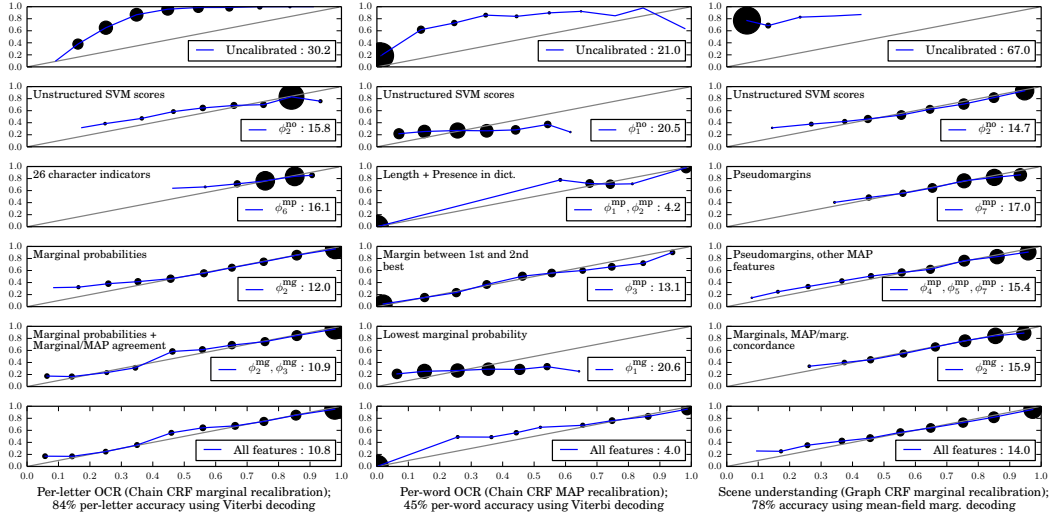

Figure 3: Feature analysis for MAP and marginal recalibration of the chain CRF (left and middle, resp.) and marginal recalibration of the graph CRF (right). Subplots show calibration curves for various groups of features from Table 1, as well as $\ell_2$ losses; dot sizes indicate relative bucket size.

Figure 2 shows that our predictions (green line) are well-calibrated in every setting. In the multiclass setting, we outperform an existing approach which individually recalibrates one-vs-all classifiers and normalizes their probability estimates [12]. This suggests that recalibrating for a specific event (e.g. the highest scoring class) is better than first estimating all the multiclass probabilities.

## 5.2 Feature Analysis

Next, we investigate the role of features. In Figure 3, we consider three structured settings, and in each setting evaluate performance using different sets of features from Table 1. From top to bottom, the subplots describe progressively more computationally demanding features. Our main takeaways are that clever inexpensive features do as well as naive expensive ones, that features may be complementary and help each other, and that recalibration allows us to add "global" features to a chain CRF. We also see that features affect only sharpness.

In the intractable graph CRF setting (Figure 3, right), we observe that pseudomarginals $\phi_7^{\mathrm{mp}}$ (which require only MAP inference) fare almost as well as true marginals $\phi_2^{\mathrm{mg}}$, although they lack resolution. Augmenting with additional MAP-based features $(\phi_4^{\mathrm{mp}}, \phi_5^{\mathrm{mp}})$ that capture whether a label is similar to its neighbors and whether it occurs elsewhere in the image resolves this.

This synergistic interaction of features appears elsewhere. On marginal chain CRF recalibration (Figure 3, left), the margin $\phi_2^{\mathrm{mg}}$ between the two best classes yields calibrated forecasts that slightly lack sharpness near zero (points with e.g. 50% and 10% confidences will have similarly small margins). Adding the MAP-marginal concordance feature $\phi_3^{\mathrm{mg}}$ improves calibration, since we can further differentiate between low and very low confidence estimates. Similarly, individual SVM and MAP-based features $\phi_2^{\mathrm{no}}, \phi_6^{\mathrm{mp}}$ (the $\phi_6^{\mathrm{mp}}$ are 26 binary indicators, one per character) are calibrated, but not very sharp. They accurately identify 70%, 80% and 90% confidence sets, which may be sufficient in practice, given that they take no additional time to compute. Adding features based on marginals $\phi_2^{\mathrm{mg}}, \phi_3^{\mathrm{mg}}$ improves sharpness.

On MAP CRF recalibration (Figure 3, middle), we see that simple features $(\phi_1^{\mathrm{mp}}, \phi_2^{\mathrm{mp}})$ can fare better than more sophisticated ones like the margin $\phi_3^{\mathrm{mp}}$ (recall that $\phi_1^{\mathrm{mp}}$ is the length of a word; $\mathcal{G}$ in $\phi_2^{\mathrm{mp}}$ encodes whether the word $y^{\mathrm{MAP}}$ is in the dictionary). This demonstrates that recalibration lets us introduce new global features beyond what's in the original CRF, which can dramatically improve calibration at no additional inferential cost.

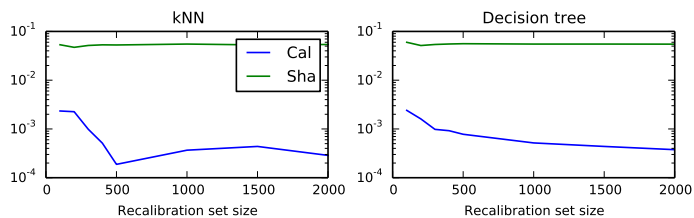

Figure 4: Calibration error (blue) and sharpness (green) of $k$-NN (left) and decision trees (right) as a function of calibration set size (chain CRF; marginal recalibration).

## 5.3 Effects of Recalibration Set Size and Recalibration Technique

Lastly, in Figure 4, we compare $k$-NN and decision trees on chain CRF marginal prediction using feature $\phi_2^{\mathrm{mg}}$. We subsample calibration sets $\mathcal{S}$ of various sizes $N$. For each $N$ and each algorithm we choose a hyperparameter (minimum leaf size for decision trees, $k$ in $k$-NN) by 10-fold cross-validation on $\mathcal{S}$. We tried values between $5$ and $500$ in increments of $5$.

Figure 4 shows that for both methods, sharpness remains constant, while the calibration error decreases with $N$ and quickly stabilizes below $10^{-3}$; this confirms that we can always recalibrate with enough data. The decrease in calibration error also indicates that cross-validation successfully finds a good model for each $N$. Finally, we found that $k$-NN fared better when using continuous features (see also right columns of Figures 2 and 3); decision trees performed much better on categorical features.

## 6 Previous Work and Discussion

Calibration and sharpness provide the conceptual basis for this work. These ideas and their connection to $l_2$ losses have been explored extensively in the statistics literature [7, 9] in connection to forecast evaluation; there exist generalizations to other losses as well [17, 10]. Calibration in the online setting is a field in itself; see [8] for a starting point. Finally, calibration has been explored extensively from a Bayesian viewpoint, starting with the seminal work of Dawid [18].

Recalibration has been mostly studied in the binary classification setting, with Platt scaling [11] and isotonic regression [13] being two popular and effective methods. Non-binary methods typically involve training one-vs-all predictors [12] and include extensions to ranking losses [19] and combinations of estimators [20]. Our generalization to structured prediction required us to develop the notion of events of interest, which even in the multiclass setting works better than estimating every class probability, and this might be useful beyond typical structured prediction problems.

Confidence estimation methods play a key role in speech recognition [21], but they require domain specific acoustic features [1]. Our approach is more general, as it applies in any graphical model (including ones where inference is intractable), uses domain-independent features, and guarantees calibrated probabilities, rather than simple scores that correlate with accuracy.

The issue of calibration arises any time one needs to assess the confidence of a prediction. Its importance has been discussed and emphasized in medicine [22], natural language processing [23], speech recognition [21], meteorology [10], econometrics [9], and psychology [24]. Unlike uncalibrated confidence measures, calibrated probabilities are formally tied to objective frequencies. They are easy to understand by users, e.g., patients undergoing diagnosis or researchers querying a probabilistic database. Moreover, modern AI systems typically consist of a pipeline of modules [23]. In this setting, calibrated probabilities are important to express uncertainty meaningfully across different (potentially third-party) modules. We hope our extension to the structured prediction setting can help make calibration more accessible and easier to apply to more complex and diverse settings.

**Acknowledgements.** This research is supported by an NSERC Canada Graduate Scholarship to the first author and a Sloan Research Fellowship to the second author.

**Reproducibility.** All code, data, and experiments for this paper are available on CodaLab at `https://www.codalab.org/worksheets/0xecc9a01cfcbc4cd6b0444a92d259a87c/`.

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
