[Reviews · NeurIPS 2015]

Submitted by Assigned_Reviewer_1

This paper proposes a method to calibrate structured predictions by minimizing the l_2 loss between true outcome and predictive probability.

The approach "re-maps" an existing structured prediction model's predictive distribution for each example to be more calibrated.

I am somewhat skeptical and confused by the motives of this work.

Calibration seems like a sensible concern for weather forecasts because the user-facing "output" of the forecast is the probability prediction itself.

Though the predictions from speech recognition, OCR, scene labeling, etc. support "user-facing" applications, the "output" is not the probability estimate for prediction, but the MAP estimate: i.e., words, labeled scenes, detected faces, etc. Should a user feel more satisfied that their structured predictor is more internally consistent (in terms of calibration) when the end-use application performance (e.g., accuracy) is worse as a result?

Can a compelling structured prediction application where calibration concerns seem justified be provided?

"More often, the probabilities obtained ... will not be useful in practice, mainly because they will not be calibrated" (line 081)

It is absurd to suggest that all of the non-calibrated structured prediction methods currently used in practice are not useful in practice.

Figure 1b: the example probabilities do not appear consistent P("land") + P("lano") < P("l***").

Much of the description of background work is light on references (in fact, there are zero references in the "Background" section!) and the connections to previous work not fully illuminated.

For example, the l_2 loss is also known as the Brier loss and has been fairly extensively studied.

The components of decomposition in (3) were originally called uncertainty, tresolution, and reliability [13].

It is unclear where e.g., the entropy term comes from -- it does not seem to match with, e.g., the Shannon entropy.

The experiments show that the calibration method proposed does indeed provide better calibration than the original structured predictions.

The small comparison between decision tree-based recalibration and k-NN-based recalibration seems insufficient.

If this idea of nearest neighbor re-calibration is novel, why not provide comparisons also in the classification setting with [15, 18, 14]?

Significantly more demonstration and/or discussion of why these previous methods cannot also be applied in some manner to these experiments is warranted.

-------------

The author feedback argues that quantifying prediction uncertainty is important, which I agree with, but this still doesn't explain why _calibrated_ uncertainty is essential.

The authors seem to equate the two, but there are many models that estimate uncertainty without being well-calibrated.

The reference provided to a structured prediction application in bioinformatics, "Taxonomic metagenome sequence assignment with structured output models," uses SVM-based methods presumably to optimize accuracy rather than calibrated uncertainty. I'm not sure how this reference is supposed to advance the argument.
Summary: Calibration methods for structured prediction are developed, but without a strong compelling use case and without comparisons to existing calibration methods.

Submitted by Assigned_Reviewer_2

Overview: In structured prediction, we are often interested not only in obtaining the most likely (e.g., the MAP) prediction, but also in a confidence assessment associated with that prediction.

Yet, even if the predictions are of high quality, the posterior probabilities assigned by the model need not be *well-calibrated*.

As an example, the model might be overly confident in its own predictions.

The theory of calibration defines two desirable properties of a probabilistic model: calibration and sharpness.

The present manuscript describes an approach for obtaining well-calibrated and sharp forecasts regarding different types of events of interest in a structured prediction model, e.g. the event that the MAP prediction is correct, or the event the a marginal prediction is correct.

Based on risk minimization theory, it is observed that a forecaster with minimal calibration error can be obtained by minimizing l2 loss on finite data.

This can be achieved using a non-parametric predictor.

In order to achieve sharpness of the forecaster, relevant features are needed. A number of such features are suggested, based, e.g., on the margin between the highest-scoring and the second highest-scoring prediction.

A number of additional strategies, such as event pooling and joint calibration are suggested to further improve the properties of forecasters.

Experiments demonstrate that the suggested approach works well.

Positive points: + The paper gives a good introduction to the concepts of calibration and sharpness.

It is clearly written. + The relevant related work is discussed in sufficient detail. + The suggested approach seems quite generally applicable and treats an important real-world problem. + The approach is well-motivated and based on established theory; its practicality is demonstrated in a number of experiments.

Negative points: - The novelty is somewhat limited, as the paper is heavily based on existing work on calibration in the binary setting, as well as previous work in speech recognition [17].

The presentation, as well as the features considered, are, however, more general than in the aforementioned work.

Overall, I still believe that the paper would be a valuable guide to practitioners seeking to obtain confidence estimates for their structured prediction models.
Summary: This is a well-written manuscript that introduces a general approach to re-calibrating the probabilities obtained from structured prediction models, for various events of interest.

The approach is based on solid theory (though largely well-known from the literature), and empirically shown to work well.

Submitted by Assigned_Reviewer_3

Overall, the paper is well-motivated and is interesting. However, it is unclear to me how the recalibrator is constructed. The experiments are pretty comprehensive. This paper presents a recalibration method for estimating the confidence of MAP and marginal

estimates for structured prediction.

Clarity: The paper is well-written, although some sections are unclear to me.

Originality: I'm not aware of other paper trying to deal with similar problem.

Significant: calibrating the structured output is an interesting and important topic.

Some questions/comments:

- It is unclear to me how the uncalibrated curve in Figure 2 is computed? Apparently the MAP score from CRF is very small, do the authors normalize it?

- It is unclear to me why the authors choose to use a decision tree model rather than a simple regression model for recalibration?

- Section 4 is unclear to me how exactly the recalibrator r is trained. Maybe I'm missing something, what is the recalibration set R in Algorithm 1.

Do the authors apply the trained model on training set to estimate T(X)? If so, will the distribution of T(X) in test time different from the estimation during the training?
Summary: Overall, the paper is well-motivated and is interesting. However, it is unclear to me how the recalibrator is constructed. The experiments are pretty comprehensive.

Submitted by Assigned_Reviewer_4

This paper proposes a method to construct calibrated output for a structured prediction system, i.e. give a probability of the output being correct. For example, if a system outputs the MAP probability of the model the calibration procedure can estimate probability for this MAP assignment being correct. The paper reuses notions of calibration and sharpness of classifiers developed for binary and multi-class classification and adopts it the case of structured prediction. The main insight of the approach consists in the fact that a regressor trained with L2 loss (features are constructed using the output of the structured prediction system and output/target is 0/1 showing whether the event is true) predicts calibrated probabilities. The paper is relatively clear, although some places are hard to understand. The studied problem is of interest and the derivations seem to be correct. The approach is an adaptation of the results for the binary and multi-class classification to the case of structured prediction. It is not always clear if the statement is a contribution or restates the prior work, but the results look very original. Comments on clarity: 1) Lines 154-156 suggest that training a binary classifier that predicts whether the event holds or not results in calibrated probabilistic output. I could not find any formal proof or citation in the support of this claim. 2) Lines 157-161 suggest training isotonic regression (in place of the binary classifier mentioned above) results in well-calibrated probabilities. It is not specified how to construct ground truth to for the isotonic regression and what algorithm to use. 3) Derivations in lines 187-190 and 238-243 are a quite quick and hard to parse. I would make sense to add more detailed derivations to the supplementary material to make text more reader-friendly. 4) Definitions of axes of calibration curves (lines 348 - 349) have typos.
Summary: This is a well-written paper targeting interesting problem. The proposed approach is original and build on the analogous studies for simpler setups.

Author Feedback
Author rebuttal: We thank the reviewers for their helpful comments.

Multiple reviewers mentioned that our work addresses a problem that is important in practice (reviewers 3, 4, 5, 6), that that our solution is novel (rev. 3, 4, 5), and that our experiments were extensive (rev. 3, 4, 5). They also voiced several valid concerns, which we address below.

1. The main concern of reviewers 1, 7 is whether probability calibration is an important problem.

In particular reviewer 1 points out that the "user-facing output" is a MAP prediction, not a probability. However, reporting the probability of being correct is often a natural thing to do, for instance in medical applications such as diagnosing the presence of infectious microbes (see PMC3131843 in PubMed for a structured prediction formulation of the problem). Even if probabilities are not seen by the user, they are useful for determining whether to report the MAP prediction at all. For example, Siri needs to determine if it has understood a user command or if it should ask for a particular type of clarification (for instance, for a specific word).

2. Reviewer 1 is concerned with a lack of comparison to previous methods.

Reviewer 1 may have two kinds of methods in mind.
(1) Previous general-purpose recalibration methods apply only in the binary or multiclass settings; extending them to structured learning involves new subtleties that we address in our paper (e.g. how to choose domain general features that are both informative and easy to compute). We also show in Figure 2a that applying our framework in the multi-class setting (a special case of structured prediction) produces improvements over an existing method.
(2) Existing techniques in speech recognition resemble our framework but involve highly domain-specific features (i.e. derived from the acoustic model, from word lattices, etc.) and do not always explicitly target calibration (e.g. some works "recalibrate" using SVMs, a notoriously uncalibrated method). See Jiang, 2005 for a survey. To our knowledge, the settings we consider have no widely accepted domain-specific calibration methods to which we can compare.

3. Other concerns of Reviewer 1

Reviewer 1 questioned the purpose of comparing kNN and random forests. We propose to use kNN with large datasets and continuous features; RFs handle better small datasets and discrete features. Our comparison was meant to illustrate this. However, our main contribution is the entire framework of events and features, not the choice of recalibrator.

We also thank reviewer 1 for pointing out that the sentence line 81 is too strong. We meant to say that the probabilities do not encode the empirical frequency of the event being correct.

Finally, we thank reviewer 1 for pointing out the lack of references in the Background section; all references were previously in the Previous Work section, but we will move them to the Background. We also thank reviewer 1 for pointing out that "uncertainty" and not "entropy" is the correct technical term (we borrowed this term from [3]); we will correct this in the next version.

4. Reviewers 5, 7 found certain parts of the paper unclear.

Reviewer 7 asks us how the margin p(y^{MAP-1}) - p(y^{MAP-2}) is computed. In the chain model, we compute y^{MAP-2} using k-best Viterbi (for k=2). In the graph model, we compute marginals q_j(x_j) using mean field (line 317), and for each j we look up the x_j with the second highest probability.

Reviewer 5 asks us how the recalibrator is constructed and whether the training and test distributions of features are identical. Note that we assume that X, Y are sampled from P; since features \phi and events E are deterministic functions, \phi_E(X,Y) and E(X) have identical training and test distributions. The recalibrator is trained to predict E from \phi on the training set; it will be calibrated on the test set by empirical risk minimization theory and because we are using the l_2 loss (which implicitly optimizes for calibration).

We also propose (line 266) that related events E_i (e.g. "was letter i decoded correctly?") may be pooled, and the recalibrator may be trained on the union of the training sets { \phi_{E_i}(X, Y), E_i(X) }. In this case, the training and test proportions of the E_i need to be similar (i.e. the user should query the probability of both types of events in the same proportions); we show in our experiments that this is a reasonable assumption to make in practice. This is indeed a confusing aspect of our method, and we thank the reviewer for pointing it out.

We will clarify all these points in the next version of the manuscript; in particular, we have already prepared a formal, mathematically rigorous discussion of event pooling.